# Functional Copy-Number Alterations as Diagnostic and Prognostic Biomarkers in Neuroendocrine Tumors

**DOI:** 10.3390/ijms25147532

**Published:** 2024-07-09

**Authors:** Hayley Vaughn, Heather Major, Evangeline Kadera, Kendall Keck, Timothy Dunham, Qining Qian, Bartley Brown, Aaron Scott, Andrew M. Bellizzi, Terry Braun, Patrick Breheny, Dawn E. Quelle, James R. Howe, Benjamin Darbro

**Affiliations:** 1Interdisciplinary Graduate Program in Genetics, University of Iowa, Iowa City, IA 52242, USA; hayley-vaughn@uiowa.edu (H.V.); terry-braun@uiowa.edu (T.B.); 2Stead Family Department of Pediatrics, University of Iowa Health Care, Iowa City, IA 52242, USA; heather-major@uiowa.edu (H.M.); evangeline-kadera@uiowa.edu (E.K.); timothy-dunham@uiowa.edu (T.D.); qining-qian@uiowa.edu (Q.Q.); 3Department of Surgery, University of Iowa Health Care, Iowa City, IA 52242, USA; kendall-keck@uiowa.edu (K.K.); aaron-scott@uiowa.edu (A.S.); james-howe@uiowa.edu (J.R.H.); 4Department of Biomedical Engineering, University of Iowa, Iowa City, IA 52242, USA; bartley-brown@uiowa.edu; 5Department of Pathology, University of Iowa, Iowa City, IA 52242, USA; andrew-bellizzi@uiowa.edu; 6Department of Biostatistics, University of Iowa, Iowa City, IA 52242, USA; patrick-breheny@uiowa.edu; 7Department of Neuroscience and Pharmacology, University of Iowa, Iowa City, IA 52242, USA; dawn-quelle@uiowa.edu

**Keywords:** neuroendocrine tumors, functional copy-number alterations, fluorescence in situ hybridization, diagnostic biomarkers, risk-stratifying biomarkers

## Abstract

Functional copy-number alterations (fCNAs) are DNA copy-number changes with concordant differential gene expression. These are less likely to be bystander genetic lesions and could serve as robust and reproducible tumor biomarkers. To identify candidate fCNAs in neuroendocrine tumors (NETs), we integrated chromosomal microarray (CMA) and RNA-seq differential gene-expression data from 31 pancreatic (pNETs) and 33 small-bowel neuroendocrine tumors (sbNETs). Tumors were resected from 47 early-disease-progression (<24 months) and 17 late-disease-progression (>24 months) patients. Candidate fCNAs that accurately differentiated these groups in this discovery cohort were then replicated using fluorescence in situ hybridization (FISH) on formalin-fixed, paraffin-embedded (FFPE) tissues in a larger validation cohort of 60 pNETs and 82 sbNETs (52 early- and 65 late-disease-progression samples). Logistic regression analysis revealed the predictive ability of these biomarkers, as well as the assay-performance metrics of sensitivity, specificity, and area under the curve. Our results indicate that copy-number changes at chromosomal loci 4p16.3, 7q31.2, 9p21.3, 17q12, 18q21.2, and 19q12 may be used as diagnostic and prognostic NET biomarkers. This involves a rapid, cost-effective approach to determine the primary tumor site for patients with metastatic liver NETs and to guide risk-stratified therapeutic decisions.

## 1. Introduction

Neuroendocrine tumors (NETs) are epithelial neuroendocrine neoplasms with high metastatic potential. Up to 60% of patients have advanced metastatic liver disease at the time of diagnosis [1,2]. Primary NETs in the abdomen are regularly missed due to small tumor size, body habitus, and limitations of abdominal imaging [1]. Twelve to fifty percent of primary gastroenteropancreatic NETs (GEP NETs) go undetected by imaging [2].

GEP NETs that originate in the pancreas (pNETs) and small bowel (sbNETs) comprise half of all GEP NETs [3]. The two tumor sites cause similar signs and symptoms, including abdominal pain, nausea, poor appetite, weight loss, and diarrhea [3]. Distinguishing between these two primary sites is important for guiding proper clinical management.

The incidence of GEP NETs is 3.56 per 100,000 people [3], but there remains a relatively high proportion of tumors that have an unknown primary site (0.86 per 100,000 people) [3]. While the incidence of sbNETs is twice as high as that of pNETs (12,000 vs. 6000 cases annually in the United States), pNETs are more aggressive, with a 5-year survival of 50% compared to 69% for sbNETs [3,4]. Treatment options for Grade 1 or Grade 2 (G1/G2) pNETs include surgery, somatostatin analogues (SSAs) [5], peptide-receptor radionuclide therapy (PRRT) [6], everolimus (an mTOR inhibitor) [7,8,9], sunitinib [10] (a VEGF inhibitor), and capecitabine/temozolomide chemotherapy [11,12]. There is limited consensus on the order of treatment when patients have progressive disease (PD). Treatment options for G1/G2 sbNETs include surgery, PRRT, and everolimus [5,6,13]. Treatment order is more established in sbNETs, with SSAs being the primary treatment, followed by PRRT and then everolimus for PD. 

Immunohistochemistry is routinely used to infer site of origin in metastatic NETs of occult origin [14,15,16]. The most commonly employed markers include CDX2 (midgut) [17], polyclonal PAX8 (pancreas) [18], TTF-1 (lung) [19], and SATB2 (rectum) [20]. None of these are NET-specific, though, and all were adapted for this purpose from more widespread use as markers of adenocarcinoma site of origin [21]. Because of the infrequency of NETs relative to other malignant epithelial tumors, complementary (serotonin for midgut) or best-in-class (islet 1 and PAX6 for pancreas and OTP for lung) NET-specific markers are not on the test menus of most clinical laboratories [15,22,23,24]. Many laboratories have switched from polyclonal to better-performing (for adenocarcinoma applications) monoclonal PAX8 antibodies, the latter of which are non-reactive in pancreatic NETs (i.e., polyclonal PAX8-positivity in pNETs is due to cross-reactivity with PAX6). Even with access to all the best-performing markers at the University of Iowa, up to 5% of NETs still defy site-of-origin assignment after exhaustive immunohistochemical and radiologic evaluation. Against this backdrop, the lead authors of the 2022 World Health Organization NET Classification stated that there remains a critical need for additional biomarkers to better classify NETs [25]. 

Copy-number alterations (CNAs) are changes in chromosomal content that result in gain or loss of. copies of DNA segments, while functional copy-number alterations (fCNAs) are changes in chromosomal content that also impact expression of genes within the region. Somatic fCNAs are predicted to be cancer drivers as opposed to bystander lesions [26]. CNAs can be detected through techniques such as whole-genome sequencing (WGS), optical genome mapping, chromosomal microarray platforms (CMA) or fluorescence in situ hybridization (FISH), while fCNAs can be detected through pairing those techniques with gene-expression analysis. Over the last decade, CNA-profile studies of NETs have yielded promising results, but notable discrepancies between studies have been observed [27,28,29,30,31,32,33,34,35,36,37]. Robust analysis of CNA profile differences between pNETs and sbNETs is still needed.

We present here the results of a pipeline used to identify differences in fCNA profiles between pNETs and sbNETs, as well as between early- and late-disease-progression tumors, with the goal of developing a panel of clinically applicable FISH probes for use in diagnostic and prognostic NET biomarker testing. This type of testing has direct relevance to metastatic NETs of unknown origin detected in the liver. To identify diagnostic NET biomarkers, a discovery cohort of patient-matched normal primary metastatic tumors underwent CMA and RNA-sequencing to identify fCNAs in 64 patients diagnosed with GEP NET (31 pNETs and 33 sbNETs). We further analyzed the prognostic value of these fCNAs in NETs, specifically with regard to time to progression. We replicated fCNAs using FISH on patient-specimen-derived tissue microarrays (TMAs) and selected probes based on their location within fCNAs of interest. Based on these collective findings, we propose a testing algorithm for primary site (pNET vs sbNET) and disease progression (late disease progression [LDP] or early disease progression [EDP]) that can be rapidly and inexpensively performed on formalin-fixed, paraffin-embedded (FFPE) NET tissue. Overall, these findings support fCNAs’ utility as effective diagnostic and prognostic biomarkers in NETs. 

## 2. Results

### 2.1. Differences in Copy-Number Alteration between pNETs and sbNETs

To establish primary and metastatic CNA profiles in pNETs and sbNETs, normal primary metastatic tumors underwent CMA profiling. In total, 47 pNET (28 primary, 19 metastatic) and 63 sbNET (32 primary, 31 metastatic) specimens were assessed by CMA. No significant CNA differences were observed between primary and metastatic tumors for either NET type (Appendix A). 

Considerable evidence suggests there are unique CNAs in pNETs and sbNETs, but discrepancies between studies have been observed [27,28,29,30,31,32]. To determine the comprehensive CNA profiles of pNETs and sbNETs, normal-tissue–primary-tumor pairs underwent CMA profiling. A total of 49 pNET and 63 sbNET patients were tested. We observed more copy-number-gain events in chromosomes 4, 5, 7, 9, 12, 13, 14, and 17–20 in pNETs (Fisher exact test *p* < 0.05). Copy-number-loss events in chromosomes 9 and 18 were detected almost exclusively in sbNETs (Fisher’s exact test *p* < 0.05). These observations indicate robust and reproducible CNA-profile differences between pancreatic and small-bowel NETs (Figure 1 and Appendix A). 

Functional copy-number alterations (fCNAs) are somatic changes in chromosomal content that affect the expression of genes within the region. To establish fCNA profiles in pNETs and sbNETs, we performed patient-matched RNA-sequencing of 31 pNETs (Appendix A) and 33 sbNETs (Appendix A) that underwent CMA profiling. To analyze the prognostic value of fCNAs in NETs, we separated our samples based on time to progression: late disease progression, or LDP, (>24 months until progression, n = 17) and early disease progression, or EDP, (<24 months until progression, n = 46). CNAs that were present in at least 15% of the cohort with significant concordant gene-expression changes (increased expression in a chromosomal gain and decreased expression in chromosomal loss) were identified as potential fCNA biomarkers (Table 1). fCNA profiles were compared, and a panel of differentiating fCNAs was selected (Table 2). 

The results showed that gains in 5q31, 7q31.1, 9p22.1, 17q11.2, 18q21.1, and 19q13.2 indicated a pancreatic primary tumor site, while a gain in 4p16.3 and a loss in 9p22.1 and/or 18q21.1 indicated a small-bowel primary site. Notable loss of chromosome 18 in sbNETs is well established in the literature [27,29,30,33,34,35,36]. Potential prognostic NET biomarkers included gains in 7q31.1, 9p22.1, and 19q13.2. 

### 2.2. FISH Validation of fCNA Differences between pNETs and sbNETs

#### 2.2.1. Diagnosis

FISH is a cytogenetic technique that can be performed on formalin-fixed, paraffin-embedded (FFPE) tissue to identify gene- and locus-specific CNAs. To replicate the evidence of the diagnostic potential of our selected fCNAs and respective probes, FISH analyses were performed on 300 nuclei of pNET (n = 60, Appendix A) and sbNET (n = 82, Appendix A) FFPE specimens. Our results indicate that CCNE1 copy-number gain is the strongest indicator of the primary site, with observed 5x higher rates in pNET samples (44% vs. 9%, Χ^2^ = 31.87, *p* < 0.00001) (Table 3). Additional copy-number gains in SMAD4 (25% vs. 5%, Χ^2^ = 15.69, *p* = 0.00008), ERBB2 (22% vs. 6%, Χ^2^ = 10.63, *p* = 0.001), and CDKN2A (22% vs. 7%, Χ^2^ = 9.07, *p* = 0.003) were observed 3–5× more frequently in pNETs tumors (Figure 2). Copy-number loss in SMAD4 was reported in 60% of sbNET tumors, which is consistent with the literature [27,29,30,33,34,35,36]. Normal copy-number status in CCNE1 was significantly more frequent in sbNET tumors (64% vs. 40%, Χ^2^ = 10.99, *p* = 0.0009). There were no significant CNA differences between pNET and sbNET tumors reported in CKS1B, FGFR3, CSF1R, and MET. 

Thresholding copy-number loss on FISH of FFPE sections is challenging, as signal is assessed in a 2D section of a 3D structure (i.e., the nucleus), resulting in the incorrect impression of loss in some subsets of normal nuclei. We assessed the background apparent deletion rate due to sectioning and found an average background deletion rate of 28.3%. We used this important value to distinguish between loss due to artifact and true copy-number loss (Appendix A).

#### 2.2.2. Prognosis

Next, we assessed whether these same fCNAs could be used as prognostic NET biomarkers. To determine if our fCNAs findings could predict disease progression, we separated our specimens into late-disease-progression (LDP) and early-disease-progression (EDP) samples with a cutoff of 5.5 years until reported disease progression. FISH analyses were performed on 300 nuclei of pNET (LDP = 24, EDP = 17) and sbNET (LDP = 41, EDP = 36) FFPE specimens. Copy-number gain in CDKN2A was 2.2× more frequent in LDP pNET tumors (42% vs. 19%, Χ^2^ = 12.48, *p* = 0.0004) (Table 4). SMAD4 copy-number gain was 2.9× more frequent in EDP pNET tumors (42% vs. 10%, Χ^2^ = 11.50, *p* = 0.0007). There were no prognostic implications for sbNET samples using these probes in our data set (Appendix A). These results validate the use of CNAs in CDKN2A and SMAD4 as independent potential prognostic pNET biomarkers.

### 2.3. Logistic Regression, Performance Metrics and Clinical Decision Tree Development

Logistic regression analysis is used to determine the probability of multiple independent variables with one dichotomous outcome. Logistic regression can be used to analyze the sensitivity, specificity, and receiver operating characteristic curve (ROC) metrics to assess the quality of a model. Using our FISH data set, each specimen per probe was given a “normalized score” (see Clinical Decision Tree Development in the Methods section) to use for logistic regression analysis (Appendix A). The results indicate MET, ERBB2, SMAD4, and CCNE1 copy-number status is significantly associated with primary tumor site (Table 5). CCNE1 had the strongest effect on primary site (0.92–1.00, CI = 0.41–1.00, *p* = 0.034), followed by SMAD4 (0.90–1.00, CI = 0.39–1.00, *p* = 0.046), ERBB2 (0.62–0.99, CI = 0.43–0.99, *p* = 0.001), and MET (0.70–0.93, CI = 0.33–0.99, *p* = 0.035). CDKN2A had a higher effect than MET on primary site, however, it did not reach significance in multivariate analysis. The performance metrics of our model are comparable to those achieved with IHC (Table 5) [14,15,16]. Based on these metrics, our model resulted in an area under the ROC curve (AUC) score of 0.902 (Figure 3).

When assessing the prognostic impact of these probes, gain of CCNE1 was associated with early disease progression (EDP) in pNETs (gain, 0.46–0.89, CI = 0.24–0.99, *p* = 0.039) and sbNETs (normal, 0.58–0.91, CI = 0.41–0.99, *p* = 0.043) (Appendix A). In addition, FGFR3 gain was associated with EDP in sbNETs (0.70–0.95, CI = 0.52–0.99, *p* = 0.0004) (Appendix A). CSF1R loss was suggestive of LDP in sbNETs (0.57, CI = 0.40–0.73, *p* = 0.07). Unlike our single-probe analysis, CDKN2A and SMAD4 had no impact on pNET prognosis in the multivariate analysis. 

We created a clinical decision tree (DT) using CNAs as diagnostic biomarkers (Figure 4). Our DT suggests assessing copy-number status for ERBB2, CCNE1, CDKN2A, and MET is effective for differentiating between pNETs and sbNETs. If ERBB2 is gained, the tumor has a 90% chance of being from the pancreas; however, if ERBB2 is lost or normal CCNE1 copy-number status should be assessed. If CCNE1 is gained, then the tumor has a greater than 80% chance of being from the pancreas but if it is lost or normal, CDKN2A copy-number status should be assessed. If CDKN2A is gained or normal, the tumor has a slightly greater than 60% chance of being from the pancreas and a 40% chance of being from the small bowel. If CDKN2A is lost and MET is either lost or normal, the sample has a greater than 80% chance of being from the small bowel; however, if MET is gained and ERBB2 is normal, it has a greater than 85% chance of being from the pancreas. Lastly, if MET is gained but ERBB2 is lost, the tumor has a greater than 90% chance of being from the small bowel. SMAD4 was not differential in our DT, which may be due to CCNE1 and SMAD4 copy number being highly correlated in our dataset (0.635) (Appendix A). Overall, these findings support the use of CNAs in MET, CDKN2A, ERBB2, and CCNE1 as simple diagnostic NET biomarkers.

## 3. Discussion

The aim of this paper was to determine if CNAs could be used as NET biomarkers by integrating patient-matched CNA and gene-expression data to identify fCNAs. We replicated fCNAs by selecting probes based on their location within fCNAs of interest using FISH on patient-specimen-derived tissue microarrays with the goal of developing a fast clinical DT that could be applied in any testing laboratory to expedite medical decision-making. 

FISH is a widely available methodology that offers a cost-effective approach with relatively rapid turnaround time. Through the identification, replication, and prioritization of FISH probes in this study, there is potential to improve the diagnostic assessment of NETs while reducing the cost and the processing time of results, thus improving patient care.

Our findings support the use of CNAs in *MET*, *ERBB2*, *SMAD4,* and *CCNE1* as diagnostic NET biomarkers and resulted in an AUC of 0.902. This compares favorably to a previously published immunochemistry-based algorithm that resulted in an ROC of 0.864 for distinguishing between tumors of pancreatic versus small-bowel or pulmonary origin [38]. As previously mentioned, IHC is currently used to infer site of origin in metastatic NETs of occult origin [14,15,16]. Taking the evidence together, we recommend assessing the copy-number status of *MET, CDKN2A, ERBB2,* and *CCNE1* to differentiate between pNETs and sbNETs. Although *SMAD4* was statistically associated with primary site, its copy-number status was highly correlated with that of *CCNE1* (0.635). The current leading NET prognostic biomarkers include tumor grade, tumor stage, and IHC analysis of genes such as p53 and ATRX/DAXX [15,23,25]. Our findings suggest that *FGFR3*, *CDKN2A*, *SMAD4,* and *CCNE1* are valuable prognostic NET biomarkers. Many studies of other tumor types show that *CCNE1* gain is a key marker of poor prognosis that may dictate more aggressive clinical management [39,40,41,42,43,44,45]. 

Limitations of our study include the moderate sample size of the dataset. This reflects the rarity of NETs, a factor that makes it challenging to gather larger cohorts, and this is especially limiting for our time-to-progression analysis. For our CMA discovery cohort, the average PFS was 2.4 years, which is different from the PFS of our FISH validation cohort and that in the literature, which is approximately 5.5 years [27,28,29,30,33]. Another limitation includes the missing values of the dataset. To combat this, we used the median score for each probe to impute the missing data; however, observed FISH results for all variables and all samples would provide more accurate overall results. A general limitation is that data gathered from clinical samples and electronic health records are not always complete across all samples. 

To our surprise, there were no significant CNA differences in *FGFR3* and *CSF1R* between pNETs and sbNETs using FISH and logistic regression analyses even though we observed significant CNA differences in our CMA-data discovery cohort. Secondly, *CKS1B* copy-number gain was not associated with PFS in this study but has been correlated with poor PFS in pan-cancer analyses [46,47,48,49,50,51,52]. It is important to note that those studies did not include assessments of the correlation between GEP NET prognosis and *CKS1B* copy-number status. Lastly, we determined there was no significant difference in *CDKN2A* copy-number loss between pNETs and sbNETs, although our CMA data suggested otherwise. *CDKN2A* has been shown to be a tumor suppressor in pNETs, and focal *CDKN2A* loss has been observed [28]. Additionally, *CDKN2A* loss might not have been observed due to the inherent challenges of assessing loss (as opposed to gain) in FFPE samples. 

In summary, we found significant CNA differences in *CDKN2A*, *ERBB2*, *SMAD4*, and *CCNE1* between pNET and sbNET tumor samples after combined analyses of chromosomal microarrays, RNA sequencing, and fluorescence in situ hybridization data. Using the FISH data, we performed logistic regression analysis and derived performance metrics while developing a clinical decision tree to help determine the primary tumor site and guide risk-stratified therapeutic decisions for metastatic tumors of unknown origin detected in the liver. This combinatorial approach to biomarker identification has proven highly effective and may represent a powerful way to define clinically relevant biomarkers for additional NET primary sites in the future. 

## 4. Materials and Methods

### 4.1. Patient Cohort

All patients in this single-institution study were enrolled under an Institutional Review Board-approved protocol. The CMA discovery cohort consisted of patient-matched normal primary metastatic tumors of 47 pNETs (28 primary, 19 metastatic) and 63 sbNETs (32 primary, 31 metastatic) (Appendix A). Normal primary metastatic tumors include the primary and metastatic tumor specimens as well as normal, healthy adjacent tissue derived from the same patient. The FISH replication cohort consisted of patient-matched normal primary metastatic tumors of 84 pNETs (76 primary, 3 metastatic liver, 5 metastatic lymph node) and 98 sbNETs (86 primary, 7 metastatic liver, 5 metastatic lymph node). Patient-cohort demographic data are listed in Appendix A. Fresh tissue samples were collected and placed in RNAlater solution (Thermo Fisher Scientific, Waltham, MA, USA), and nucleic acids were isolated using the RNeasy Plus Universal Mini Kit (Qiagen, Valencia, CA, USA) or DNeasy blood & Tissue Kit (Qiagen, Valencia, CA, USA) per the protocols recommended by the manufacturers.

### 4.2. Microarray Protocols 

Microarray experiments were performed using the NimbleGen Human CGH 720 K Whole-Genome Tiling version 3.0 array (Roche NimbleGen; Madison, WI, USA) and the Affymetrix CytoScan High-Definition (HD) array (Affymetrix array, Santa Clara, CA, USA) according to the manufacturers’ instructions. Calculation of log2 ratio values and quality-control metrics were assessed using the NimbleScan software tool (version 2.5; Roche NimbleGen) or the ChAS (Chromosome Analysis Software) tool (version 1.1.2; Affymetrix) (CytoScan). CNA calling and data interpretation were performed using the Nexus Copy Number software (version 6.1, BioDiscovery; El Segundo, CA, USA) and the rank segmentation algorithm (for Nimblegen arrays) or the SNPRank segmentation algorithm (CytoScanHD) supplied with the Nexus software suite (version 2.5; Roche NimbleGen) [53]. Allele-specific copy-number analysis of tumors (ASCAT) was performed to identify ploidy and percent normal tissue. 

### 4.3. RNA Processing

RNA-seq was performed within the Genomics Division of the University of Iowa Institute of Human Genetics (University of Iowa, Iowa City, IA, USA) using the Illumina TruSeq protocol (Illumina, Inc., San Diego, CA, USA), as previously described [54]. RNA-seq count data were normalized using FPKM. To identify differentially expressed genes, TopHat (v 2.1.0, John Hopkins University, Baltimore, MA, USA), Cuffquant, Cuffnorm, and Cuffduff were performed, comparing tumors to healthy adjacent pancreatic or small-bowel tissue. Statistically significant expression change was determined by assessing the false discovery rate (FDR) adjusted *p*-value (q-value), with significance defined as q < 0.05. 

### 4.4. fCNA Identification

The R package (v 3.4.4) iGC [55] integrates sample-paired copy-number and gene-expression analysis to identify concordant differential gene expression. Log2 copy-number values from NimbleGen (Roche NimbleGen; Madison, WI, USA) or Affymetrix (Affymetrix array, Santa Clara, CA, USA) were utilized to analyze the association between copy number and mRNA levels of 31 pNET and 33 sbNET samples. CNAs that were observed in at least 20% of samples were assessed. Allelic-imbalance and loss-of-heterozygosity lesions were removed. Genes for which expression was predicted to be driven by either copy-number gain (and increased expression) or copy-number loss (and decreased expression) events and which met previously defined *p* and FDR thresholds were retained. No fold-change threshold was utilized for the iGC analysis. 

### 4.5. Fluorescence In Situ Hybridization (FISH) 

TMAs were assembled from formalin-fixed, paraffin-embedded lesions of 60 pNET and 82 sbNET primary and metastatic specimens arrayed in triplicate (2-micron sections). FISH studies were performed using Empire Genomics FISH probes *CKS1B*-20-OR, *FGFR3*-20-OR, *CSFR1*-20-GR, *MET*-20-GR, *CDKN2A*-20-GR, *ERBB2*-20-OR, *SMAD4*-20-GR, and *CCNE1*-20-OR (Empire Genomics, Inc., New York, NY, USA) per the protocol recommended by the manufacturer (empiregenomics.com/resources/protocols-procedures). One hundred nuclei per section and three sections per patient specimen, for a total of three hundred nuclei, were scored. Briefly, slides were baked for a minimum of 8 h at 45 °C, deparaffinized with Hemo-De, and then rehydrated before they were heated in Dako pre-treatment 20×. The samples were digested with pepsin at 37 °C for 5 min. After digestion, slides were washed with Dako buffer and dehydrated with a series of 70%, 85%, and 100% ethanol. The slides were then probed, denatured, and hybridized for 48 h at 37 °C. Unbound probe was washed from the slides, and they were counterstained with DAPI. FISH slides were analyzed at 100× magnification using CytoVision (Leica Microsystems, Wetzlar, Germany) filters. 

Counts for loss-, normal-, and gain-signal-pattern nuclei were average for each specimen and probe set. Statistical differences were assessed using the Chi-Square test with significance defined as *p* < 0.05. Firth’s bias-reduced logistic regression was used to estimate the predicted probability of association of CNAs with a given tissue of origin. Copy-number loss was defined as occurrence observed in a proportion less than or equal to 28.3% of nuclei showing fewer than two copies to account for “pseudo-loss” due to the plane of sectioning. Copy-number gain was defined as occurrence observed in a proportion greater than or equal to 15.3% of nuclei. These values were calculated by assessing the average percent copy-number loss, normal copy-number status, and copy-number gain within our entire dataset and selecting the greatest-variable-range value for pNETs (loss) and sbNETs (gain), respectively. Duplicate CNAs of the same probe set with the largest variable ranges were chosen. Samples outside of these loss and gain thresholds were determined to be normal. Descriptive statistics for copy-number loss, normal copy-number status, and copy-number gain are listed in the table provided (Appendix A). 

### 4.6. Logistic Regression Analysis and DT Development

Logistic regression and statistical analyses were performed by assessing the 8 biomarker-locus variables in the FISH dataset. To do so, each biomarker in each sample was given a score. The score was determined by calculating the percent normal nuclei, subtracting the percent loss nuclei, and adding the percent gain nuclei. Therefore, a sample probed with a biomarker that resulted in 60% normal, 28% loss, and 12% gain nuclei would be given a score of 44 (60 − 28 + 12). Loss, normal, and gain values were determined by thresholds set based on the original variables. Missing data were imputed with the median score for each probe. The biomarker and response variables were tested within a pipeline with the Analysis of Overdispersed Data (aod- version 1.3.2) and ggplot2 (version 3.4.4) R (v 4.3.1) packages. 

DT dataset curation was performed by assessing the 8 biomarker-locus variables in the FISH dataset categorically. For example, if a probe for any given sample was lost, a value of 1 was designated, while normal and gains were assigned values of 2 and 3, respectively. Missing data points were assigned a value of 0. The decision tree was developed using the rpart (version 4.1.23) and partykit (version 1.2.20) R (v 4.3.1) packages. 

## Figures and Tables

**Figure 1 ijms-25-07532-f001:**
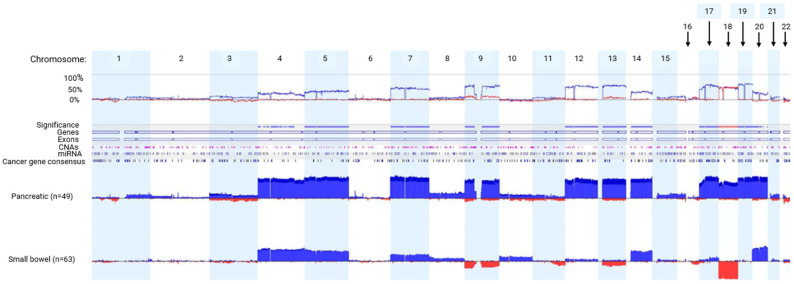
CMA data comparing pNET and sbNET tumor samples. The top row indicates chromosomes 1–22. Below is the frequency of the CNA events. Significance indicated by blue significance line. Genes, exons, CNAs, microRNAS (miRNAs), and cancer-related genes are listed below the significance bar. Blue represents copy-number gain. Red represents copy-number loss.

**Figure 2 ijms-25-07532-f002:**
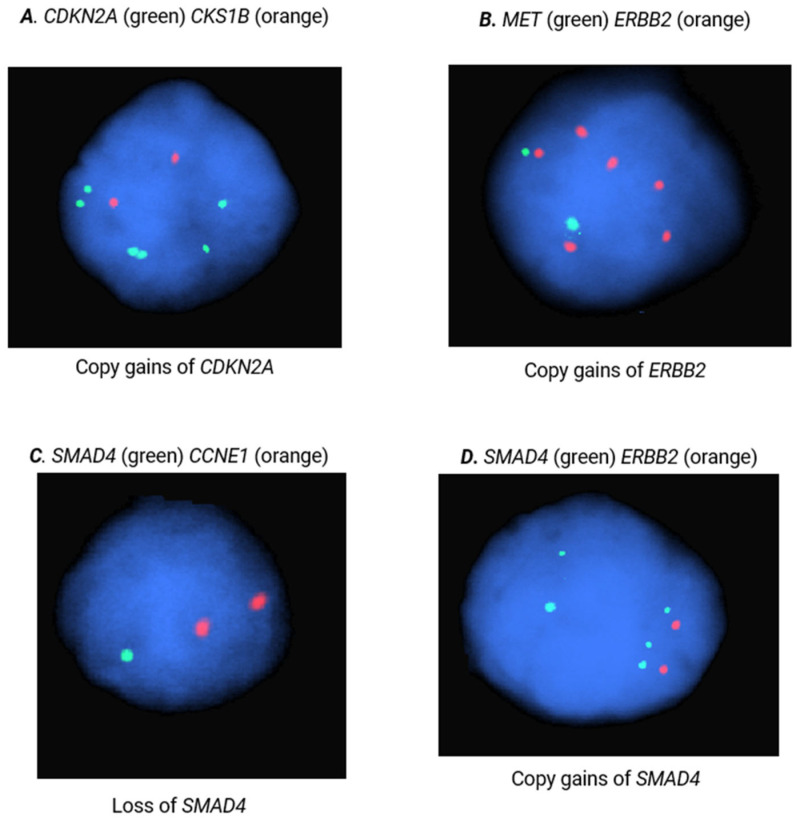
FISH analysis of CKS1B, MET, CDKN2A, ERBB2, SMAD4, and CCNE1. Images are representative of copy gains of (**A**) CDKN2A, (**B**) ERBB2, and (**C**) SMAD4, as well as loss of (**D**) SMAD4.

**Figure 3 ijms-25-07532-f003:**
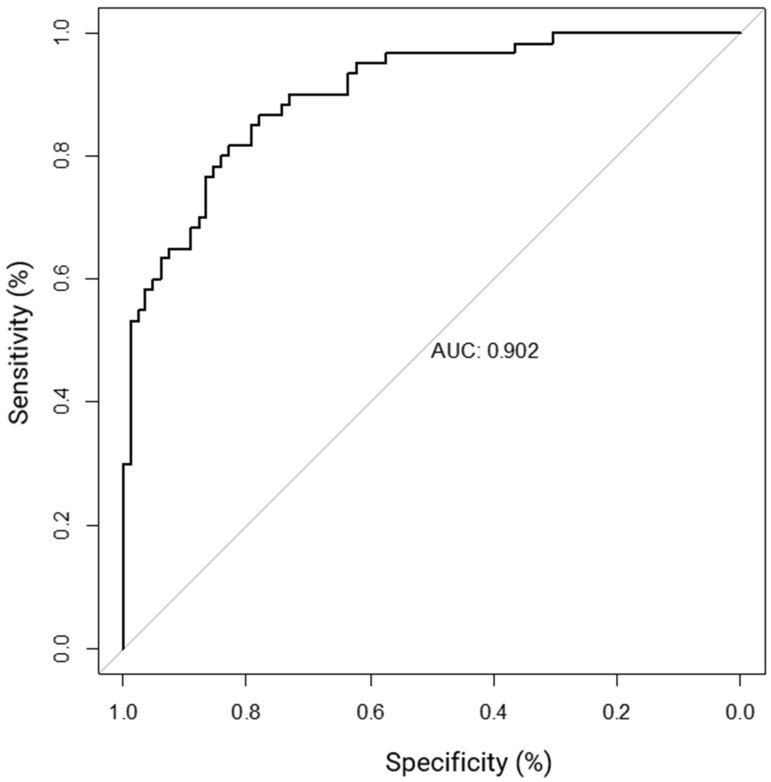
Receiver operating characteristic curve (ROC) curve analysis using CNAs in CCNE1, SMAD4, ERBB2, and CDKN2A as diagnostic NET biomarkers.

**Figure 4 ijms-25-07532-f004:**
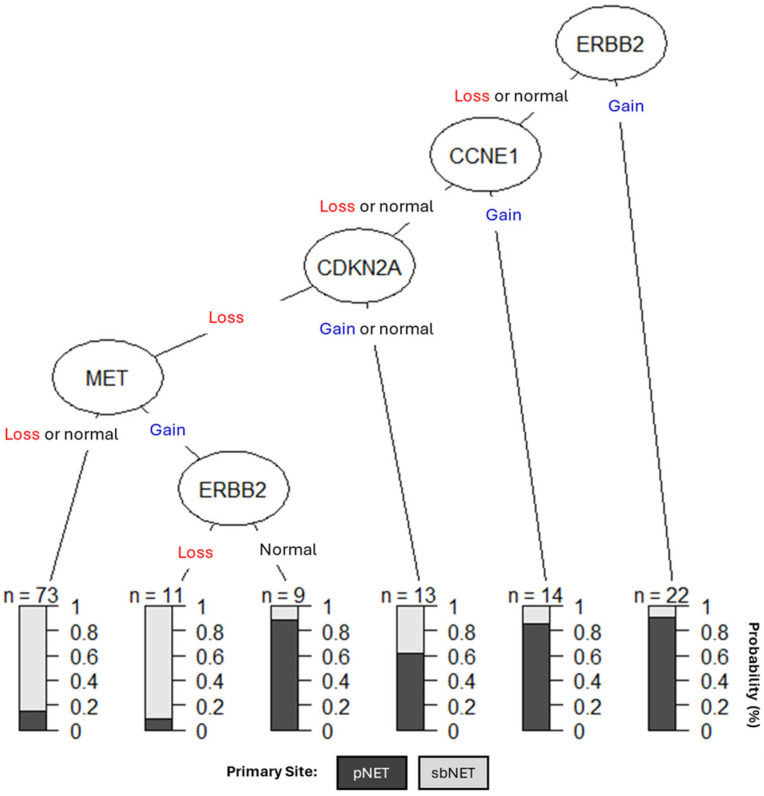
Clinical DT models using CNAs in ERBB2, CCNE1, CDKN2A, and MET as NET biomarkers. The shading of the boxes represents the most likely primary tumor site, while the numbers on the right-hand side represent the predicted probabilities by site.

**Table 1 ijms-25-07532-t001:** fCNA loci and examples of genes within those loci that exhibited copy-number alterations and concordant gene-expression changes.

			CNA (%)			RNA-Seq (Log2Fold)			
fCNA Locus & *Genes*	*p*	FDR	L	N	G	L	N	G	Δ N vs L or G
4p16.3									
*family with sequence similarity 218 (FAM218A)*	6.04 × 10^−6^	0.01	0%	48%	52%	NA	0.7	3.7	3.0
*UNC-5 Netrin Receptor C (UNC5C)*	0.0002	0.03	0%	42%	58%	NA	1.3	4.0	2.7
5q21.2-31.3									
*CART Prepropeptide (CARTPT)*	7.98 × 10^−5^	0.03	0%	35%	65%	NA	0.6	6.4	5.8
*Protocadherin Gamma Subfamily B, 3 (PCDHGB3)*	0.0002	0.03	0%	42%	58%	NA	0.6	3.1	2.5
7q21.3-q31.1									
*ATP-Binding Cassette Subfamily B MemberB (ABCB1)*	0.0004	0.04	0%	39%	61%	NA	1.6	5.2	3.6
*Reelin (RELN)*	0.0004	0.04	0%	35%	65%	NA	0.8	4.7	3.9
9p24.2-q34.11									
*Very-Low-Density Lipoprotein Receptor (VLDLR)*	0.0001	0.03	3%	42%	55%	−0.1	−0.4	1.3	1.6
*Torsin Family 1 Member B (TOR1B)*	0.0004	0.04	3%	39%	58%	0.6	0.8	1.7	0.9
17p11.2-13.1 17q11.2–15.3									
*TNF Receptor-Associated Factor 4 (TRAF4)*	0.0001	0.03	3%	26%	71%	−0.2	−0.4	1.3	1.7
*Ras-Related Dexamethasone Induced 1 (RASD1)*	0.0003	0.03	6%	42%	52%	4.5	1.1	4.5	3.0
18q21.1									
*Ectopic P Granules 5 Autophagy Tethering Factor (EPG5)*	0.0002	0.01	64%	36%	0%	−0.6	0.2	NA	−0.9
*Ferrochelatase (FECH)*	0.001	0.03	64%	36%	0%	−0.3	0.3	NA	−0.6
19q13.12-13.42									
*RNA Polymerase II Subunit I (POLR2I)*	0.0001	0.03	0%	32%	68%	NA	1.0	3.1	2.0
*Leucine-Rich Repeat and Fibronectin Type III Domain Containing 1 (LRFN1)*	7.27 × 10^−5^	0.03	0%	32%	68%	NA	−0.7	1.6	2.3

L = loss. N = normal. G = gain. Δ = difference. *p*-value, false discovery rate (FDR).

**Table 2 ijms-25-07532-t002:** Differential panel of CNAs and associated commercially available and clinically used cancer-related gene probes.

fCNA Results	Biomarker	Commercially Available and Clinically Used Probes (Locus)
1q21.3+	EDP	*CKS1B* (1q21.3)
4p16.3+	LDP	*FGFR3* (4p16.3)
5q31.2-31.3+	EDP	*CSF1R* (5q32)
7q21.3-q31.1+	pNET/EDP	*MET* (7q31.2)
9p24.2-q34.11+	pNET	*CDKN2A* (9p21.3)
17p11.2-17q13.1+	pNET	*ERBB2* (17q12)
17q11.2-17q15.3+	EDP	
18q21.1-	sbNET	*SMAD4* (18q21.2)
19q13.12-19q13.42+	pNET	*CCNE1* (19q12)

EDP = early disease progression. LDP = late disease progression. + indicates copy-number gain. - indicates copy-number loss.

**Table 3 ijms-25-07532-t003:** FISH results assessing the average percentage of copy-number loss, normal copy-number status, and gain of biomarkers between pNETs and sbNETs. Χ^2^ = Chi-square.

Probe	CN Loss (%)	CN Normal (%)	CN Gain (%)
*CKS1B*	Χ^2^ = 2.59 *p*-value (*p*) = 0.27		
pNET (n = 27)	36	54	9
sbNET (n = 26)	27	59	14
Χ^2^	1.88	0.4	1.17
*p*	0.17	0.53	0.28
*FGFR3*	Χ^2^ = 7.04 *p* = 0.70		
pNET (n = 16)	26	51	23
sbNET (n = 44)	21	55	24
Χ^2^	0.7	0.32	0.03
*p*	0.4	0.57	0.87
*CSF1R*	Χ^2^ = 4.63 *p* = 0.10		
pNET (n = 21)	17	56	27
sbNET (n = 31)	23	62	15
Χ^2^	1.13	0.74	4.34
*p*	0.29	0.39	0.04
*MET*	Χ^2^ = 5.84 *p* = 0.054		
pNET (n = 42)	24	53	24
sbNET (n = 53)	28	62	11
Χ^2^	0.42	1.64	5.84
*p*	0.52	0.2	0.02
*CDKN2A*	Χ^2^ = 9.25 *p* = 0.01		
pNET (n = 19)	22	56	22
sbNET (n = 14)	29	64	7
Χ^2^	1.29	1.33	9.07
*p*	0.26	0.25	0.003
*ERBB2*	Χ^2^ = 10.77 *p* = 0.005		
pNET (n = 37)	22	56	22
sbNET (n = 31)	29	65	6
Χ^2^	1.29	1.69	10.63
*p*	0.26	0.19	0.001
*SMAD4*	Χ^2^ = 29.12 *p* < 0.00001		
pNET (n = 22)	26	49	25
sbNET (n = 17)	60	35	5
Χ^2^	23.58	4.02	15.69
*p*	<0.00001	0.04	0.00008
*CCNE1*	Χ^2^ = 21.92 *p* 0.00001		
pNET (n = 22)	16	40	44
sbNET (n = 17)	28	64	9
Χ^2^	4.04	10.99	31.87
*p*	0.04	0.0009	<0.00001

CN = copy number. *p*-value calculated by Chi-square test (Χ^2^). False discovery rate (FDR) = 0.006.

**Table 4 ijms-25-07532-t004:** FISH results assessing the average percentage of copy-number loss, normal copy-number status, and gain of biomarkers between LDP and EDP.

Probe	CN Loss (%)	CN Normal (%)	CN Gain (%)
** *CKS1B* **	Χ^2^ = 1.81 *p* = 0.41		
LDP (n = 6)	36	58	7
EDP (n = 10)	44	48	8
Χ^2^	1.45	1.79	0.08
*p*	0.23	0.18	0.77
** *FGFR3* **	Χ^2^ = 36.55 *p* < 0.00001		
LDP (n = 15)	24	52	24
EDP (n = 1)	63	32	4
Χ^2^	31.76	7.9	16.39
*p*	0.00001	0.005	0.00005
** *CSF1R* **	Χ^2^ = 1.41 *p* = 0.49		
LDP (n = 13)	19	56	24
EDP (n = 6)	13	61	26
Χ^2^	1.41	0.4	0.08
*p*	0.23	0.53	0.78
** *MET* **	Χ^2^ = 0.09 *p* = 0.96		
LDP (n = 16)	24	49	27
EDP (n = 11)	23	52	27
Χ^2^	0.06	0.08	0.01
*p*	0.81	0.78	0.93
** *CDKN2A* **	Χ^2^ = 12.49 *p* = 0.002		
LDP (n = 6)	16	42	42
EDP (n = 3)	23	58	19
Χ^2^	1.56	5.12	12.48
*p*	0.21	0.02	0.0004
** *ERBB2* **	Χ^2^ = 2.80 *p* = 0.25		
LDP (n = 15)	20	52	28
EDP (n = 10)	27	54	19
Χ^2^	1.36	0.08	2.25
*p*	0.24	0.78	0.13
** *SMAD4* **	Χ^2^ = 12.78 *p* = 0.002		
LDP (n = 5)	37	53	10
EDP (n = 14)	23	48	29
Χ^2^	4.67	3.02	11.5
*p*	0.03	0.08	0.0007
** *CCNE1* **	Χ^2^ = 0.52 *p* = 0.77		
LDP (n = 5)	15	39	46
EDP (n = 14)	17	42	41
Χ^2^	0.15	0.19	0.51
*p*	0.7	0.67	0.48

CN = copy number. LDP = late disease progression. EDP = early disease progression. *p*-value calculated by Chi-square test (Χ^2^). False discovery rate (FDR) = 0.006.

**Table 5 ijms-25-07532-t005:** Logistic regression results based on primary tumor site. (a) Reported scores of statistical analyses per probe. (**b**) Quality control metrics for the overall model with reported AUC.

(a.) Probe	Score	Effect Size	95% CI	z Value	*p*-Value
*CKS1B*	−40 to −10	0.72–0.88	0.07–1.00	−1.19	0.232
*FGFR3*	−30–2	0.59–0.74	0.15–0.98	−1.25	0.212
*CSF1R*	90	0.70	0.30–0.92	1.91	0.056
*MET*	80	0.70	0.33–0.91	2.11	0.035
*CDKN2A*	70–80	0.77–0.88	0.23–1.00	1.61	0.107
*ERBB2*	60–80	0.62–0.95	0.43–0.99	3.45	0.001
*SMAD4*	50–90	0.90–1.00	0.39–1.00	2.00	0.046
**(b.) Performance Metrics**					
PseudoR2			0.44		
*p*-value			5.23 × 10^−15^		
Sensitivity			0.68		
Specificity			0.89		
AUC			0.902		

CI = confidence interval. AUC = area under the ROC curve.

## Data Availability

Data are contained within the article and Appendix A.

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
