# Peer review of "Functional Copy-Number Alterations as Diagnostic and Prognostic Biomarkers in Neuroendocrine Tumors"

_ijms, 2024, doi:10.3390/ijms25147532_

Round 1

Reviewer 1 Report

Comments and Suggestions for Authors

In the present manuscript, authors demonstrated that copy number changes at chromosomal loci 4p16.3, 7q31.2, 9p21.3, 17q12, 18q21.2, and 19q12 may be used as diagnostic and prognostic NET biomarkers. This involves a rapid, cost-effective approach to determine primary tumor site for patients with metastatic liver NETs and to guide risk stratified therapeutic decisions.

Manusctipt is well written and presented data are potentially useful for scientific community. I have only a few comments to improove the overall quality:

-          Histolological pictures of the studied cases should be also presented

-          Discussion section should address these two questions:

1)      What are the currently known prognostic biomarkers of Net?

Is immunohistochemistry capable to differentiate and suggest the primary origin 

Author Response

Comment 1:  Histolological pictures of the studied cases should be also presented

Response 1: Great suggestion. We have added Supp figure 3 to fulfill this. 

Comment 2 (1): Discussion should address the current prognostic NET biomarkers.

Response 2 (1): Thank you for pointing this out. We have added to lines 286-288 "The current leading NET prognostic biomarkers include tumor grade, stage, and IHC analysis of genes such as p53 and ATRX/DAXX (CITE).”

Comment 2 (2): Discussion should address if immunohistochemistry is capable of differenting primary site. 

Response 2 (2): Excellent suggestion. We have added to lines 282-283: “As previously mentioned, IHC is currently used to infer site of origin in metastatic NETs of occult origin [14-16].”

Reviewer 2 Report

Comments and Suggestions for Authors

Manuscript entitled "Functional Copy Number Alterations as Diagnostic and Prognostic Biomarkers in Neuroendocrine Tumors"

1. The authors should list the recurrently amplified/deleted regions and the genes located within those regions in a table.

2. The authors should identify those genes with focal amplification (instead of whole chromosome copy number change) and demonstrate their fold change in figures.

3. For FISH study, the centromeres, but not other genes should be used as a reference. 

4. The authors should check TCGA data for comparison and further discussion.

5. HE for selected cases, and IHC for selected genes and proteins should be presented. 

Comments on the Quality of English Language

Acceptable

Author Response

Comment 1: The authors should list the recurrently amplified/deleted regions and the genes located within those regions in a table.

Response 1: Excellent suggestion. We have added Supplementary table S1, which details recurrent changes observed. Because we identified large chromosomal changes, adding genes that are within those regions would result in a rather large list. 

Comment 2: The authors should identify those genes with focal amplification (instead of whole chromosome copy number change) and demonstrate their fold change in figures.

Response 2: Thank you for the feedback. We did not identify any focal amplifications. 

Comment 3: For FISH study, the centromeres, but not other genes should be used as a reference. 

Response 3: Thank you for the comment. We were looking for absolute copy number differences not changes in reference to the centromere or other control loci.  

Comment 4: The authors should check TCGA data for comparison and further discussion.

Response 4: Great suggestion! Unfortunately, when we checked TCGA that data was not available.  

Comment 5: HE for selected cases, and IHC for selected genes and proteins should be presented. 

Response 5: Thank you for this suggestion. We have added Supp figure 3 to fulfill this. 

Round 2

Reviewer 2 Report

Comments and Suggestions for Authors

The revision is acceptable.

Comments on the Quality of English Language

Acceptable.